# Symmetric Space Learning for Combinatorial Generalization

## Abstract

Symmetries in representations within generative models play essential roles in predicting unobserved combinations of semantic changes, known as combinatorial generalization tasks. However, existing methods primarily focus on learning symmetries from training data, leaving the extension of trained symmetries to unseen samples unaddressed. A promising approach to addressing this limitation is leveraging geometric information on manifolds containing semantic structures for unseen data, though this remains insufficient for robust symmetry learning. In this paper, we tackle the problem of *symmetry generalization* by enforcing *symmetric space* on the latent space, leveraging semantic structures from both symmetry and manifold perspectives. We identify an equivariance-based constraint that restricts symmetry generalization and prove that: 1) enforcing the homogeneous space property of symmetric space onto the data manifold resolves this constraint, 2) a homogeneous latent manifold induces a homogeneous data manifold through diffeomorphic mappings, and 3) the isometry property of symmetric space extends local symmetries across the space. To implement this, we propose a method to align sampled points from symmetric space with their explicitly trained geodesics. We validate our approach through a detailed analysis on a toy dataset and demonstrate its effectiveness in enhancing combinatorial generalization on common benchmarks. This work represents the first effort to integrate symmetric space learning into generative models for combinatorial generalization.

## 1 Introduction

Generalizing a model to unobserved combinations of semantic factors learned during training has been a critical challenge for achieving human-like generalization (Fodor & Pylyshyn, 1988). This problem, referred to as combinatorial generalization in representation learning (Vankov & Bowers, 2019), aims to capture data structures that encode semantic relations within latent representations to enable generalization. However, most existing approaches to representation learning struggle to achieve this goal effectively (Schott et al., 2021).

Symmetry, which characterizes transformations within representations, is a crucial property for addressing the challenge of combinatorial generalization. In Higgins et al. (2022), symmetry learning has been shown to effectively capture structural information within representations, while Hwang et al. (2023) demonstrates that symmetry learning improves combinatorial generalization. However, these symmetry-based methods primarily focus on learning symmetries from observed data, limiting their ability to generalize to unseen cases.

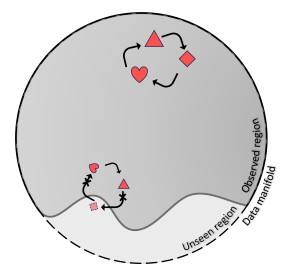

Figure 1: Failure of transferring symmetry structure onto unseen region of data manifold. Since trained group actions should be closed on the observed region, their corresponding symmetries can not affect the unseen region.

Geometric information provides a potential pathway for generalizing symmetries to unseen data. For example, a manifold represents the geometric region where samples are likely to be observed (Narayanan & Mitter, 2010; Bordt et al., 2023), and its structure can be learned through

representation learning (Bengio et al., 2013). The shortest path between two points on the manifold, known as a geodesic, helps infer relationships between unseen and seen points (Shao et al., 2018). Furthermore, traversals on the manifold of latent vectors capture factorized semantic changes in corresponding data points (Choi et al., 2021). However, these insights into manifolds and geometric information have not yet been integrated with symmetry-based approaches, leaving a gap in leveraging geometric structures for combinatorial generalization.

This paper addresses the challenge of *symmetry generalization*, which involves extending learned symmetries from observed data to unseen data in combinatorial generalization tasks. To tackle this, we propose a framework that enforces the latent vector space manifold to exhibit properties of a *symmetric space*. We propose enforcing the latent vector space as a symmetric space and provide theoretical insights into its effectiveness: 1) enforcing homogeneity on the data manifold addresses constraints that limit symmetry learning, 2) equivariant diffeomorphic mappings ensure consistency between the latent and data manifolds, and 3) symmetries in the latent space can propagate to unseen regions through isometric properties. These theoretical foundations are not stand-alone but serve as guiding principles for the practical implementation of our framework.

To make symmetry generalization computationally feasible, we introduce a sampling-based approach to induce symmetric space structures. This approach involves: 1) Generating explicit curves that connect sampled latent vectors to shared *anchor*, 2) approximating these curves as geodesics, and 3) aligning extrapolated points from reflection symmetries with the geodesic approximations. This method avoids the computational intractability of processing the entire space and enables the practical application of symmetric space learning to complex datasets.

We verify our proofs and method in Morpho-MNIST (Castro et al., 2019) and analyze its impact quantitatively and qualitatively on dSprites (Matthey et al., 2017) and 3D shapes (Burgess & Kim, 2018) datasets.

Our main contributions are summarized as follows:

1. We address the challenge of extending learned symmetries from observed to unseen data, a key bottleneck in combinatorial generalization tasks, by formalizing the problem of *symmetry generalization*.

2. We propose a practical framework, *symmetric space learning*, which bridges symmetry learning and manifold geometry to enhance generalization beyond the training data.

3. We develop a computationally efficient sampling-based approach to induce symmetric space structures by leveraging geodesic symmetries in the latent space.

4. We demonstrate the effectiveness of our method through experiments on a toy dataset (Morpho-MNIST) and its strong performance on standard benchmarks (dSprites and 3D Shapes), assessed both quantitatively and qualitatively.

## 2 BACKGROUND

A *manifold*, a mathematical structure locally equivalent to Euclidean space, is a foundational concept in machine learning due to the *manifold hypothesis* (Narayanan & Mitter, 2010), which posits that high-dimensional data lies on a low-dimensional manifold. To apply differentiable functions—integral to most machine learning methods—to the manifold must possess a *differentiable structure*. In our discussion, we focus on *Riemannian manifolds*, defined as follows (do Carmo, 1992):

**Definition 1** (Riemannian manifold). *A **Riemannian metric** on a differentiable manifold $\mathcal{M}$ is an inner product $\langle \cdot, \cdot \rangle_p$ (a symmetric, bilinear, positive-definite form) on the tangent space $T_p\mathcal{M}$ at each point $p \in \mathcal{M}$. A differentiable manifold $\mathcal{M}$ equipped with a Riemannian metric is called a Riemannian manifold.*

To accurately represent the latent manifold while preserving the structure of the data manifold, we require a mapping that preserves geometric properties, termed a *diffeomorphism*:

**Definition 2** (Diffeomorphism). *Let $\mathcal{M}$ and $\mathcal{N}$ be smooth manifolds. A map $f : \mathcal{M} \to \mathcal{N}$ is called a **diffeomorphism** if it is bijective, continuous, differentiable, and its inverse $f^{-1}$ is also differentiable.*

Symmetries play a central role in our method, and a *homogeneous space* provides a formal way to describe them on a manifold:

**Definition 3** (Homogeneous space). *A Riemannian manifold $\mathcal{M}$ is called a* **homogeneous space** *if a group $G$ acts transitively on $\mathcal{M}$*

On a homogeneous space, every point corresponds to an element of the group $G$ acting on the manifold (do Carmo, 1992). This property allows us to analyze the structure of the manifold through symmetries independent of specific coordinate systems.

To further refine the concept of symmetries on manifolds, we consider *symmetric spaces*, a special case of homogeneous spaces with additional symmetry properties:

**Definition 4** (Symmetric space). *Let $\mathcal{M}$ be a manifold, and let $p \in \mathcal{M}$ be an arbitrary point. Assume that a curve $\gamma : I \to \mathcal{M}$, where $I \subset \mathbb{R}$ is a real interval, is a geodesic with $\gamma(0) = p$. A* **geodesic symmetry** *is a map $f : U \to U$, where $U$ is a neighborhood of $p$, such that $f(\gamma(t)) = \gamma(-t)$ for all $t \in I$. If geodesic symmetries are isometries and extend globally across the manifold, then $\mathcal{M}$ is called a symmetric space (Helgason, 2001).*

## 3 SYMMETRY GENERALIZATION THROUGH SYMMETRIC SPACE

### 3.1 LIMITED EXTENSION OF TRAINED SYMMETRIES TO UNSEEN REGION

Symmetry-based machine learning methods aim to train models that capture variations in data through symmetries. Group actions, supported by well-established mathematical and physical theories, provide a concrete framework for representing these symmetries. In particular, equivariant group actions have become central to embedding symmetry properties into latent space representations. For instance, Hwang et al. (2023) proposed a method to generate novel samples by applying symmetries learned from observed data. However, this approach relies on group actions derived solely from the training data, limiting the generalization of symmetries to unseen data. We term this challenge *symmetry generalization*, which is formalized in the following proposition.

**Proposition 1** (Limit of Symmetry Generalization). *Let $X$ be a dataset and $X' \subset X$ be a subset of partially observed samples. If $G$ is a symmetry group that acts transitively on $X$, then there exists no $g \in G$ such that $g \cdot x = x'$ for $x \in X \setminus X'$ and $x' \in X'$.*

While this limitation follows directly from the definition of group actions, it highlights a critical gap in existing approaches to symmetry learning: the inability to extend symmetries beyond the training dataset. This constraint underscores the need for methods capable of generalizing symmetries to unseen data, a focus of our work.

### 3.2 SYMMETRIC SPACE INDUCTION

**Motivation: Homogeneity on Entire Data Manifold via Symmetric Space on Latent Manifold**
From a geometric perspective, the data manifold encodes semantic structures for both observed (trained) samples and potential unseen samples that share the same marginal distribution as the training data. Aligning symmetries to the manifold enables the propagation of learned symmetries to unseen data, addressing the limitations of symmetry generalization highlighted in Proposition 1. We propose enforcing the *homogeneous space* property on the data manifold to achieve this alignment. The transitivity of homogeneous spaces provides a systematic solution to Proposition 1 by ensuring that symmetries learned from training data can be extended to the entire manifold. However, directly enforcing a homogeneous structure on the data manifold is computationally intractable due to its unknown structure. To overcome this challenge, we adopt a two-step approach: 1) inducing homogeneity on the data manifold via its latent representation, and 2) extending trained local symmetries to unseen regions by leveraging the *symmetric space* property. In the subsequent sections, we formalize these steps and provide the necessary propositions to support our approach.

**Conditions for Homogeneity Transfer from Latent to Data Manifold** Existing methods in symmetry-based machine learning focus primarily on group structures, often overlooking the role of geometric structures. We need the modified condition, equivariant mapping with group action, to fit the given situation, defining a homogeneous data manifold from a latent homogeneous manifold.

In particular, the mapping between manifolds must be equivariant while preserving the geometric structure of the data manifold.

To formalize this, we establish the following conditions for a model to enable the homogeneous space framework: Let $\mathcal{D}$ be the data space and $\mathcal{H}$ be the latent space. Let $\mathcal{M} \subset \mathcal{D}$ be the data manifold and $\mathcal{N} \subset \mathcal{H}$ be the latent manifold. The model must satisfy:

1. **Homogeneity of the Data Manifold**: A group $G$ acts transitively on $\mathcal{M}$, i.e., $\mathcal{M}$ is a homogeneous space under $G$.

2. **Equivariant Diffeomorphism**: A map $\phi : \mathcal{M} \to \mathcal{N}$ is an equivariant diffeomorphism with respect to the group $G$.

The first condition extends the group action from the dataset to the entire data manifold, while the second ensures that the equivariance mapping preserves the geometric structure of the manifolds. With these conditions satisfied, we establish the following proposition to enable the induction of homogeneity on the data manifold via the latent manifold:

**Proposition 2** (Homogeneity Transfer via Equivariance). *Let $\mathcal{M}$ and $\mathcal{N}$ be manifolds, and let $\phi : \mathcal{M} \to \mathcal{N}$ be an equivariant diffeomorphism with respect to the group $G$. If $\mathcal{M}$ is a homogeneous space under the action of $G$, then $\mathcal{N}$ is also a homogeneous space under $G$, and vice versa.*

*Proof.* If $\mathcal{M}$ is homogeneous, then for any $p, q \in \mathcal{M}$, there exists $g \in G$ such that $g \cdot p = q$. By equivariance, $g \cdot \phi(p) = \phi(g \cdot p) = \phi(q)$, so $G$ acts transitively on $\mathcal{N}$. Conversely, if $\mathcal{N}$ is homogeneous, then for any $p, q \in \mathcal{M}$, we have $g \cdot \phi(p) = \phi(q)$. Since $\phi$ is a diffeomorphism, $g \cdot p = q$. Hence, $G$ acts transitively on $\mathcal{M}$. $\qquad\square$

This proposition highlights that homogeneity on the latent manifold can be transferred to the data manifold via an equivariant diffeomorphism. Although the result is straightforward, it provides a foundational step for identifying group structures that are homogeneous with the data manifold by constructing a homogeneous latent manifold.

**Homogeneity Extension via Symmetric Space Induction** Proposition 2 resolves the symmetry generalization limitation identified in Proposition 1 by enforcing homogeneity in the latent space. However, directly identifying a homogeneous group for the entire data space without additional information is impractical. To address this, we propose leveraging the *symmetric space* structure, which extends homogeneity by incorporating isometric geodesic symmetries across all point pairs on the manifold. This extension ensures symmetry generalization, as shown in the following proposition:

**Proposition 3** (Latent Space Symmetry Generalization). *Let $\phi : \mathcal{M} \to \mathcal{N}$ be an equivariant diffeomorphism from the data manifold $\mathcal{M}$ to the latent manifold $\mathcal{N}$. Suppose $\mathcal{M}$ is partitioned into observed data $X$ and unseen data $X'$, with corresponding latent partitions $Z = \phi(X)$ and $Z' = \phi(X')$. If $\mathcal{N}$ is a symmetric space $G/H_a$, and the model learns $\mathrm{Aut}(Z)$, the automorphisms of $Z$, then for any $x' \in X'$, there exist $g \in G$ and $\alpha \in \mathrm{Aut}(Z)$ such that*

$$x' = \phi^{-1}(g \cdot \alpha(a)),$$

*where $a$ is the origin point of $\mathcal{N}$.*

*Proof.* By Proposition 2, $\mathcal{M}$ is homogeneous under $G$, so for any $x \in X$ and $x' \in X'$, there exists $g \in G$ such that $g \cdot x = x'$. Let $p = \phi(x) \in Z$. Since $\mathcal{N}$ is a symmetric space $G/H_a$, and the model learns $\mathrm{Aut}(Z)$, there exists $\alpha \in \mathrm{Aut}(Z)$ such that $\alpha(a) = p$. Combining these, $g \cdot p \in Z'$, so $x' = \phi^{-1}(g \cdot \alpha(a))$, completing the proof. $\qquad\square$

This proposition establishes that trained local symmetries can be extended to unseen regions through the symmetric space structure of the latent manifold.

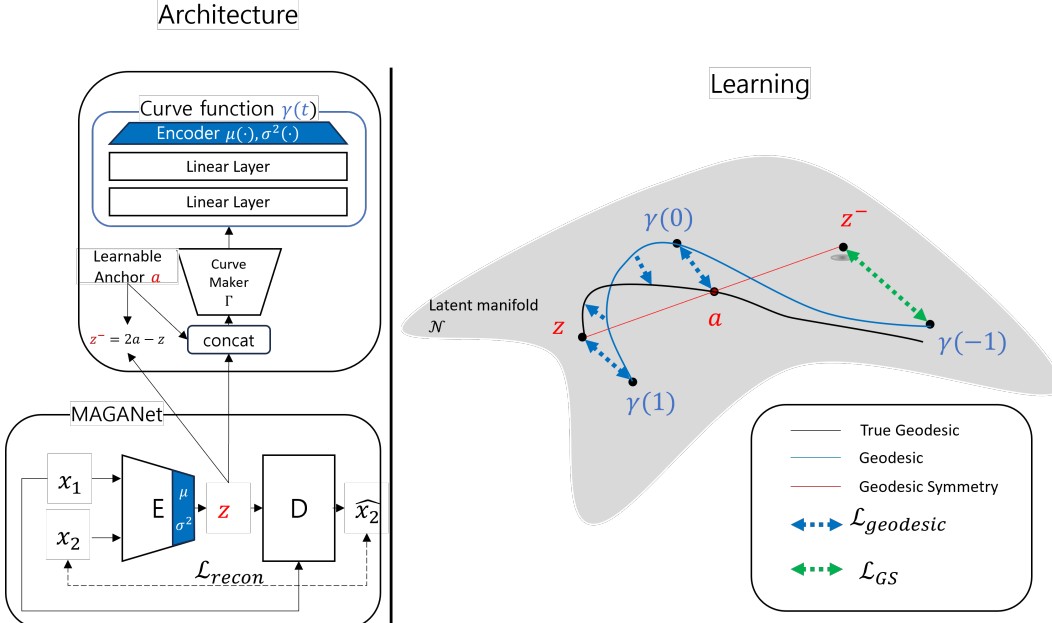

Figure 2: Overview of the network architecture on the MAGANet (left) and the process (right) of symmetry generalization: 1) aligning the curve ($\gamma(\cdot)$) to a geodesic between a training sample $z$ and anchor $a$ on symmetry space, 2) extending the alignment to an extrapolated sample $z^-$ potentially including an unseen region on the space.

## 4 METHOD

**Overview** In this section, we propose a practical method to implement symmetric space learning. This method addresses how to satisfy the equivariant diffeomorphism condition in a machine learning framework and how to effectively learn geodesic symmetries.

Figure 2 illustrates the key components of our approach, which include: 1) a network module that generates a *curve function* for a latent vector from a training sample, 2) a loss function designed to approximate this curve to a geodesic connecting the latent vector and a learnable anchor, and 3) a loss function for inducing geodesic symmetries by aligning extrapolated unseen samples through reflection. Detailed explanations of each component follow.

### 4.1 FRAMEWORK FOR CONDITIONS OF HOMOGENEITY TRANSFER

**Base Model: MAGANet** Our base model must exhibit equivariance and diffeomorphic properties to satisfy the conditions discussed in Section 3.2. We adopt MAGANet (Hwang et al., 2023) as the baseline model, which meets these requirements through its design. MAGANet combines a Variational Autoencoder (VAE) (Kingma & Welling, 2013) with a flow-based generative model, GLOW (Kingma & Dhariwal, 2018), to achieve equivariant and differentiable mapping.

The GLOW module is particularly suitable for this task because it comprises invertible and differentiable layers, ensuring bijective mappings between spaces. As demonstrated in Zhen et al. (2021), flow-based generative models can implicitly capture manifold structures, making them apt for modeling the equivariant and diffeomorphic mapping from the data manifold to the latent manifold. In our approach, the GLOW module is trained to encode the manifold structure implicitly while satisfying homogeneous group action.

**Addressing the Limitation of Local Symmetry** While the VAE encoder in MAGANet effectively models local symmetries, it does not ensure their extension to unseen regions, as noted in Proposition 1. The modeled symmetries correspond to points on the latent manifold via group ac-

tion, with the pivot acting as the manifold's origin point. However, extending this symmetry across the entire latent space requires additional mechanisms.

To complete the homogeneity of the latent manifold, we incorporate a module that enforces global structure through geodesic symmetry learning, as described in the subsequent sections.

## 4.2 Implementation for Homogeneity Extension

Building on the MAGANet model (Hwang et al., 2023), we establish an equivariant diffeomorphism between the data manifold and the latent manifold, as outlined in Proposition 3. This map ensures that the latent manifold can serve as a symmetric space to generalize symmetries learned from observed data to unseen data.

To implement symmetric space properties on the latent manifold, we design a novel module consisting of two key components: the *curve function maker* and the *anchor*. The *curve function maker* generates a curve parameterized by multi-layer perceptrons (MLPs), connecting a given latent vector to the anchor. Its objective is to approximate the curve to a geodesic, ensuring minimal distortion of the manifold structure.

Geodesic symmetries are modeled as linear extrapolations, with the additional objective of aligning the extrapolated points with the extended geodesic. In this context, the encoder of MAGANet interprets group actions as vector operations (e.g., vector subtraction). By enforcing that the linearly extrapolated points lie on the latent manifold, our method effectively extends symmetries to unseen regions, enabling symmetry generalization across the entire latent space.

**Curve Function Maker**  A starting and end point must be specified to define a geodesic on a manifold. In our implementation, the starting point is designated as the latent point $z$, while the endpoint is defined as a learnable parameter in the latent space, referred to as the *anchor* $a$. For simplicity, all geodesics are constructed to start from $z$ and end at the shared endpoint $a$. The anchor $a$ is implemented using a learnable parameter class from the PyTorch library.

To approximate a geodesic, we design a *curve function maker* $\Gamma$, parameterized by multi-layer perceptrons (MLPs), which generates the parameters of the curve function $\gamma$:

$$\Gamma(z, a) = (W_1, W_2, b_1, b_2), \tag{1}$$

where $W_1 \in \mathbb{R}^{|B| \times 1 \times l}, W_2 \in \mathbb{R}^{|B| \times l \times h}, b_1 \in \mathbb{R}^{|B| \times 1 \times l}, b_2 \in \mathbb{R}^{|B| \times 1 \times h}$. Here, $|B|$ denotes the mini-batch size, $l$ is the hidden representation dimension, and $h$ is the hidden dimension of the VAE encoder layer in MAGANet.

The curve function $\gamma$ estimates an arbitrary curve between two points as follows:

$$\gamma(t) = \mu(w) + \epsilon \cdot \sigma(w), \tag{2}$$
$$w = \text{bmm}(\text{ReLU}(\text{bmm}(t, W_1) + b_1), W_2) + b_2, \tag{3}$$

where bmm is batched matrix multiplication, $t \in \mathbb{R}^{|B| \times 1 \times 1}$, and $\mu$ and $\sigma$ are layers from the VAE encoder in MAGANet used to compute the mean and standard deviation.

Sharing components of the VAE encoder ensures the output of $\gamma$ to lie on the manifold learned by the base model, preserving the geometric structure of the latent space.

**Geodesic Approximation**  To utilize the generated curve as a geodesic, it must represent a locally shortest path along the manifold. Given the computational cost of computing exact geodesics, we adopt a discrete approximation method following (Yang et al., 2018):

$$E_{\text{apprx}} = \sum_{i=0}^{T} \left\{ (\mu(w_i) + \mu(w_{i+1}))^2 + (\sigma^2(w_i) + \sigma^2(w_{i+1})) \right\}, \tag{4}$$
$$w_i = \text{bmm}(\text{ReLU}(\text{bmm}(i/T, W_1) + b_1), W_2) + b_2, \tag{5}$$

where $T$ represents the number of uniformly separated intervals along the curve. This approximation allows the model to compute geodesics for each latent variable in batch during every training step.

To ensure that the curve function $\gamma$ starts at the latent point $z$ and ends at the anchor $a$, we enforce boundary conditions during training. The *geodesic loss* is then defined to optimize the curve function maker $\Gamma$ and the anchor $a$ as follows:

$$\mathcal{L}_{\text{geod}} = l_1(z, \gamma(\mathbf{1})) + l_1(a, \gamma(\mathbf{0})) + E_{\text{apprx}}, \tag{6}$$

where $l_1$ denotes the L1 loss. This loss function ensures that the generated curve adheres to the properties of a geodesic while maintaining the starting and ending conditions.

**Learning Geodesic Symmetries**   The base model encoder is designed to model group actions, which we extend to geodesic symmetries in our approach. The model learns to align each reversed point along the geodesic with its linear reflection point, denoted by $z^- = 2a - z$, where $z$ is the latent vector and $a$ is the anchor to facilitate the identification of reverse points through geodesic symmetry.

While reflection points generally do not lie on the manifold, the curve function approximates geodesics constrained to the manifold. The alignment procedure minimizes the gap between the reflection points and the geodesics on the manifold, ensuring that the learned symmetries better represent the geometric structure of the latent space. To achieve this, the model minimizes the discrepancy between the symmetry learned by the model and the symmetry generated by geodesic symmetry using the following L1 loss:

$$\mathcal{L}_{\text{gs}} = l_1(z^-, \gamma(-1)). \tag{7}$$

Here, $\gamma(-1)$ represents the point extrapolated along the geodesic beyond the anchor in the direction opposite to $z$.

The total loss for training combines the base model's loss and the additional losses for geodesic approximation and geodesic symmetries:

$$\mathcal{L} = \mathcal{L}_{\text{Base}} + \zeta(\mathcal{L}_{\text{geod}} + \mathcal{L}_{\text{gs}}), \tag{8}$$

where $\zeta$ is a hyperparameter to balance the contributions of the geodesic loss $\mathcal{L}_{\text{geod}}$ and the geodesic symmetry loss $\mathcal{L}_{\text{gs}}$, while $\mathcal{L}_{\text{Base}}$ is defined in Equation 12.

Further implementation details, including specific architectures and hyperparameter settings, are provided in Appendix C.

## 5   Related Works

**Symmetry Representation**   Symmetry and its group representation play a foundational role in representation learning (Higgins et al., 2022). Many approaches emphasize the connection between disentangled representations and group structures to enhance performance (Higgins et al., 2018), improving disentanglement results (Cha & Thiyagalingam, 2023; Yang et al., 2022; Zhu et al., 2021a). However, these methods primarily learn symmetries from observed data, limiting their applicability to unseen cases. In contrast, our method extends symmetry learning to unobserved data, addressing a critical gap in existing literature.

**Geometry in Generative Models**   A geometric perspective has been shown to improve representation learning in generative models. For instance, Fumero et al. (2021) demonstrated that manifold learning enhances disentanglement in latent spaces, while Falorsi et al. (2018) explored manifold learning integrated with group actions. These studies suggest that combining geometry and symmetry benefits representation learning. Manifold learning has also been used for generalization tasks, such as Out-of-Distribution robustness in NLP (Ng et al., 2020) and improved classification performance (Vural & Guillemot, 2016). Building on these insights, we propose leveraging manifold structures to enhance generative generalization, particularly for unseen combinations of semantic factors.

**Combinatorial Generalization**   Combinatorial generalization has emerged as a critical challenge in machine learning (Vankov & Bowers, 2019). Symbolic representations and disentangled representations have been proposed as solutions (Vankov & Bowers, 2019; Montero et al., 2020). However, disentangled representations often fail to generalize to unseen data, as shown by Montero et al.

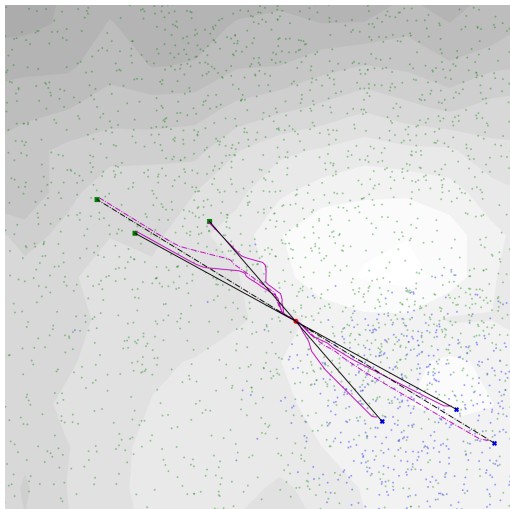
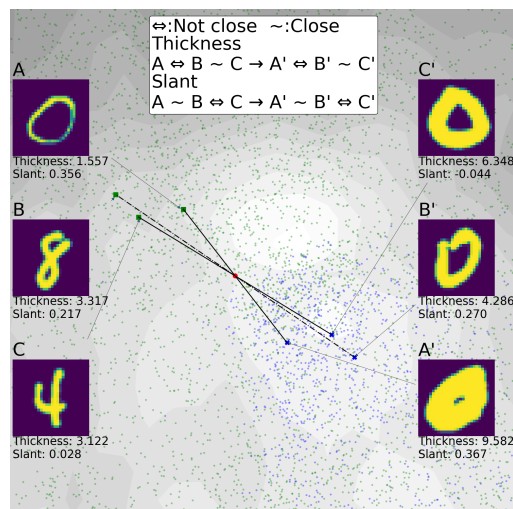

(a) Approximated Geodesic on Latent Manifold    (b) Local Structure Transfer on Geodesic Symmetries

Figure 3: 2D latent space visualization of VAE trained with geodesic symmetry. (a) Approximated geodesic (magenta line) between test samples (blue crosses) and anchor (red dot), and its extension to reflection points (green squares), ranging from $(-0.3, -0.1)$ to $(0.3, 0.5)$. (b) Trained geodesic symmetries ranging from $(-0.4, -0.3)$ to $(0.6, 0.6)$. The images on the left side are sampled test data, and those on the right are the nearest training data from the reflected points. Each image is captioned with its deformation information. The logarithm of the Riemannian volume form determines the contours. The brighter region indicates a higher likelihood of being on the manifold.

(2022). Recent advancements explore alternative approaches, such as sufficient conditions for generalization (Wiedemer et al., 2023) and architectures modeling group actions (Hwang et al., 2023). Our work builds on these ideas by introducing symmetric space learning, which incorporates group actions and latent geometry to overcome the limitations of representation failure and restricted symmetry domains.

## 6 EXPERIMENTS

### 6.1 IN-DEPTH MANIFOLD AND SYMMETRY ANALYSIS ON MORPHO-MNIST

To evaluate the effects of our method, we analyze the geodesic and local structure on the latent manifold of models trained on the Morpho-MNIST dataset (Castro et al., 2019). The test set comprises images of the thick digit zero, while the training set includes all other digits and thickness level combinations. We construct a VAE model with a curve function generator trained to approximate geodesics on the latent manifold. Detailed experimental settings are provided in Appendix D.1.

As shown in Fig. 3a, the approximated geodesic lines traverse regions of higher likelihood, indicated by brighter areas. This result demonstrates that the proposed curve function generator effectively minimizes distances on the latent manifold, producing meaningful geodesics. Moreover, these geodesics connect samples to their corresponding reflection points via the anchor, as intended.

Additionally, Fig. 3b illustrates test samples alongside their nearest training samples of reflected points. Notably, the test samples appear to be transformations of the training data, with the slant factor preserved (although not explicitly labeled) and the thickness factor altered. Specifically, the thickness of the transformed samples shows reduced variance compared to other transformed samples, as observed in pairs of nearby training samples (displayed in the middle and bottom rows).

These findings indicate that: 1) Transformations along the geodesic modify certain features while preserving others. 2) Local structural information is effectively maintained through geodesic symmetries.

Table 1: Binary Cross Entropy($\downarrow$) in Combinatorial Generalization. (underbar: better result in comparison with MAGANet, bold: the best across all models.)

| | Symmetry | Model | R2E | | | R2R | | |
| --- | --- | --- | --- | --- | --- | --- | --- | --- |
| | | | Case1 | Case2 | Case3 | Case1 | Case2 | Case3 |
| dSprites | X | VAE | 9.23 | **8.36** | 14.59 | 255.11 | 248.65 | 47.93 |
| | | β-VAE (β = 2) | 14.69 | 14.22 | 23.07 | 261.14 | 206.62 | 60.31 |
| | | β-VAE (β = 4) | 23.72 | 25.26 | 31.08 | 182.88 | 169.37 | 144.83 |
| | | β-VAE (β = 8) | 56.15 | 52.15 | 62.56 | 123.62 | 189.15 | 179.86 |
| | O | CLGVAE | 9.69 | 9.34 | 16.24 | 448.28 | 343.98 | 66.04 |
| | | MAGANet | 9.83 | 10.01 | 16.99 | 75.20 | 84.77 | 19.75 |
| | | MAGANet+GS (ours) | 9.07 | 9.89 | **14.34** | **50.05** | **68.04** | **17.27** |
| 3D Shapes | X | VAE | 3920.50 | 3605.59 | 3919.35 | 3794.83 | 3755.13 | 3709.96 |
| | | β-VAE (β = 2) | 3915.29 | 3610.20 | 3929.23 | 3794.28 | 3746.58 | 3709.95 |
| | | β-VAE (β = 4) | 3927.78 | 3615.51 | 3926.88 | 3802.02 | 3759.98 | 3717.60 |
| | | β-VAE (β = 8) | 3946.45 | 3630.99 | 3933.02 | 3849.37 | 3770.86 | 3733.34 |
| | O | CLGVAE | 7428.74 | 3624.50 | 3952.53 | 3862.72 | 3766.28 | 3740.66 |
| | | MAGANet | **3905.28** | **3590.13** | 3900.96 | 3795.04 | 3721.75 | 3966.20 |
| | | MAGANet+GS (ours) | 3911.31 | 3592.77 | **3899.77** | **3774.06** | **3716.48** | **3698.88** |

## 6.2 Performance on Combinatorial Generalization Benchmarks

**Dataset**   We conduct experiments on two benchmark datasets commonly used for evaluating combinatorial generalization: dSprites (Matthey et al., 2017) and 3D Shapes (Burgess & Kim, 2018). Following Montero et al. (2020), each dataset is split into training and test sets under two settings: *Recombination-to-Elements* (R2E) and *Recombination-to-Range* (R2R). Unlike previous works, we exclude three specific combinations in each setting to minimize the influence of arbitrary factor selection.

The dSprites dataset consists of binary images with a white shape on a black background, uniquely defined by five factors: shape, size, orientation, x-position, and y-position. The 3D Shapes dataset comprises images of objects within a scene and is uniquely determined by six factors: object shape, object scale, object orientation, object hue, wall hue, and floor hue.

In the R2E setting, we exclude three combinations of specific values or small ranges for each factor in both datasets. In the R2R setting, we exclude three combinations of a specific value and a range for each factor. Further details on dataset settings and excluded combinations are provided in Appendix D.2.

**Model and Training Settings**   We evaluate our method using MAGANet (Hwang et al., 2023), as described in 4.1. For comparisons, we include VAE (Kingma & Welling, 2013) and β-VAE (Higgins et al., 2016), which do not explicitly learn symmetries, and Commutative Lie Group VAE (CLGVAE) (Zhu et al., 2021b), which does explicitly learn symmetries.

All models use a latent dimension of 10. Training is conducted with a batch size of 128, a learning rate of 0.001, and 100 epochs. For the dSprites dataset, models are trained using binary cross-entropy loss, while mean-squared error loss is used for the 3D Shapes dataset. The energy function interval $T$ in Equation 5 is set to 16, enabling efficient geodesic approximation.

For β-VAEs, we test three values of $\beta$ (2, 4, and 8). For MAGANet and the GS-equipped version, hyperparameters $\beta_{kl}$ and $\beta_{lr}$ are fixed at 1, based on optimal values determined through grid search. The scaling factor $\zeta$ for our proposed loss is set to 1 across both datasets. Additionally, MAGANet-based models require a pivot for inference; we select the median sample from the training dataset as the pivot for both datasets.

**Quantitative Analysis**   Table 1 summarizes the binary cross-entropy results for combinatorial generalization tasks on dSprites and 3D Shapes datasets under R2E and R2R settings. Our method, MAGANet+GS, achieves consistent and significant improvements across most metrics. In the dSprites dataset, while R2E performance in Case2 is slightly lower than VAE, our method demonstrates notable improvements in R2R, notably achieving the best results in Case1 and Case2. For the 3D Shapes dataset, MAGANet+GS consistently outperforms the baseline models, including MAGANet, in all cases for both R2E and R2R settings. These results emphasize the effectiveness of our approach in extending symmetries to unseen data, as indicated by the substantial performance gains

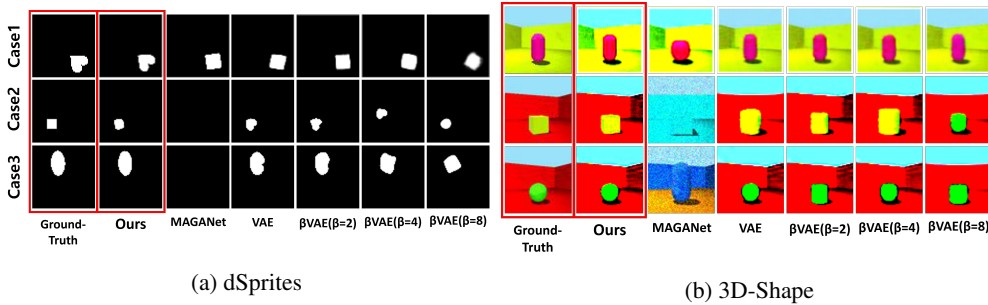

(a) dSprites

(b) 3D-Shape

Figure 4: Generated Images in Recombination-to-Range (R2R) Setting

over MAGANet. The consistent improvements highlight the robustness and versatility of our method in addressing combinatorial generalization.

**Qualitative Analysis** Fig. 4a and Fig. 4b showcase the generated images for the R2R setting across different datasets and models. For VAEs, the generated outputs often appear blurry or distorted blobs, mainly when $\beta$ values are higher in the dSprites dataset. In contrast, our method effectively captures critical factors, such as the distinct shapes of objects, where baseline models struggle to generalize to unseen cases. For example, in the first row of Fig. 4a, our method successfully generates heart shapes, while other methods produce squares, ellipses, or amorphous blobs. Similarly, in the second row of Fig. 4b, VAEs generate cylinders instead of cubes, indicating their inability to capture the underlying factors of variation. Overall, our approach demonstrates robust generalization capabilities in generating unseen combinations and preserving geometric and semantic fidelity.

**Ablation Study for Geodesic loss and GS loss** To validate the effectiveness of our proposed losses, including the geodesic loss and the geodesic symmetry loss, we evaluated performance under three scenarios: 1) training with all proposed losses, 2) training without the geodesic symmetry loss, and 3) training with the geodesic symmetry loss on a curve but without the geodesic approximation loss. The experiments were conducted using the same training and testing settings as the main experiments

Table 2: Ablation study to verify effeteness of proposed loss.

| Method | R2E | R2R |
|---|---|---|
| Ours | 9.07 | **50.05** |
| $(-)$ GS loss | 8.86 | 65.03 |
| $(-)$ geodesic | **8.49** | 68.05 |

on the dSprites dataset (Case 1), with the $\zeta$ hyperparameter fixed at a constant value of 1. As shown in Table 2, the R2R setting achieves the best results when all components of our method are included, while excluding any loss slightly decreases the performance in the R2E setting. These findings suggest that our method effectively uncovers broad unseen portions of the data.

# 7 CONCLUSION

In this paper, we addressed the problem of trained symmetries being limited in their applicability to unobserved data for combinatorial generalization. We demonstrated that structuring the latent vector space as a symmetric space enables the generalization of trained symmetries and proposed a novel method to induce symmetric space by generating specific samples and aligning them to the approximated geodesic. The effectiveness of our approach was validated through in-depth analysis of tests on the toy Morpho-MNIST dataset and further corroborated by experiments on widely used benchmarks, including dSprites and 3D Shapes, for combinatorial generalization tasks. Our work is the first to establish the utility of integrating manifold and symmetry learning to enhance combinatorial generalization. This contribution opens up promising directions for future research, such as exploring diverse sampling strategies tailored to specific data characteristics and extending the approach to tackle a broader range of generalization tasks beyond combinatorial generalization.

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

## A  BROADER IMPACT AND LIMITATION

### A.1  BROADER IMPACT

This paper addresses a mathematical approach to enhancing the generalization capabilities of generative models. Such improved generalization capabilities have the potential to aid in the creation of novel artifacts, enabling more individuals to easily create what they desire. Moreover, the mathematical methodologies proposed in this paper can be applied to a wide range of machine learning tasks and models.

### A.2  LIMITATION

The proposed method is based on the manifold hypothesis; thus, if the given dataset does not conform to this hypothesis, our method may be less effective. Additionally, for practical implementation, we assume that the base model does not sample outlier variances in the context of variational inference. Therefore, if such outliers are present, our model cannot transform the latent manifold into a fully symmetric space but only into a locally symmetric one.

Since geodesics on the manifolds are based on variance-based metric as the Mahalanobis metric (Chadebec & Allassonniere, 2022), the geodesic symmetry can induce isometry when involved points have the small difference of variance values. However, more direct methods to inject the isometry may be more effective to induce symmetric space to wider cases.

## B  DETAIL ON THEORETICAL BACKGROUND

**Group Action**  Group $(G, *)$ is a mathematical structure which is tuple of a set $G$ and binary operation $*$ closed on the set. Group should satisfy following axioms:

1. **(associativity)** $a * (b * c) = (a * b) * c$

2. **(identity element)** there exists $e \in G$ such that $a * e = e * a$

3. **(inverse element)** there exists $a^{-1} \in G$ such that $a * a^{-1} = a^{-1} * a = e$

for every $a, b, c \in G$. Group play as an atom of representing symmetry. The group action on a set $X$ of a group $G$ is a map $f : G \times X \to X$ which satisfy following axioms:

1. **(identity)** $f(e, x) = x$

2. **(compatibility)** $f(g, f(h, x)) = f(gh, x)$

for an identity $e \in G$ and every $g, h \in G$ and every $x \in X$. We can decompose natural phenomena into objects and symmetries of those via group action.

**Definition 5.** *Let $G$ be a group and $X$ be a $G$-space. The action is said to be a transitive if there exists $g \in G$ such that $g * x = y$ for any $x_1, x_2 \in X$.*

This means that every point $x \in X$ can be translated into any point in $X$ with an action $g \in G$.

**Geometry** In geometry and topology, manifold means topological space that locally resembles to Euclidean space at every point on it. More formally, smooth manifold which is type of manifold can be defined as follows (do Carmo, 1992).

**Definition 6.** *A smooth (or differentiable) manifold of dimension $n$ is a set $\mathcal{M}$ and a family of injective mappings $x_\alpha : U_\alpha \subset \mathbb{R}^n \to \mathcal{M}$ of open sets $U_\alpha$ of $\mathbb{R}^n$ into $\mathcal{M}$ such that*

    *1. $\bigcup_\alpha x_\alpha(U_\alpha) = \mathcal{M}$.*

    *2. for any fair $\alpha, \beta$ with $x_\alpha(U_\alpha) \cap x_\beta(U_\beta) = W \neq \emptyset$, the sets $x_\alpha^{-1}(W)$ and $x_\beta^{-1}(W)$ are open sets in $\mathbb{R}^n$ and the mappings $x_\beta^{-1} \circ x_\alpha$ are differentiable.*

    *3. The family $\{(U_\alpha, x_\alpha)\}$ is maximal relative to the conditions (1) and (2).*

Every point on manifold have tangent space which is the vector space tangent to manifold.

**Definition 7.** *Let $\mathcal{M}$ be a differentiable manifold. A differentiable function $\alpha : (-\epsilon, \epsilon) \to \mathcal{M}$ is called a (differentiable) curve in $\mathcal{M}$. Suppose that $\alpha(0) = p \in \mathcal{M}$, and let $D$ be the set of functions on $\mathcal{M}$ that are differentiable at $p$. The tangent vector to the curve $\alpha$ at $t = 0$ is a function $\alpha' : D \to \mathbb{R}$ given by*

$$\alpha'(0)f = \left.\frac{d(f \circ \alpha)}{dt}\right|_{t=0}, f. \tag{9}$$

*A tangent vector at $p$ is the tangent vector at $t = 0$ of some curve $\alpha : (-\epsilon, \epsilon) \to \mathcal{M}$ with $\alpha(0) = p$. The set of all tangent vectors to $\mathcal{M}$ at $p$ will be indicated by tangent space $T_p\mathcal{M}$.*

We can think a *open subset of manifold* $\mathcal{M}$ at $p \in \mathcal{M}$ and it is informal definition of *neighborhood* of $p$.

## C ARCHITECTURES

### C.1 BASELINES

To implement $\beta$-VAE (Higgins et al., 2016), we use the structure introduced in (Burgess et al., 2018). The encoder consist of four convolution layers with 32 channels, two fully connected layer with 256 nodes and fully connected layer with $d$ nodes where $d$ is latent vector dimension, and the decoder consist of transpose of encoder structure. ReLU activation is used for each layer except last layer of the encoder and the decoder.

To implement MAGANet (Hwang et al., 2023), we followed proposed architecture. The encoder for modeling group actions is same with the VAE encoder architecture. The decoder consist of a linear layer with out bias to apply group action and GLOW model (Kingma & Dhariwal, 2018). The GLOW model has three flow modules and each flow module has three flow block and squeeze layer. Each flow block is composed of ActNorm, $1 \times 1$ convolution without LU decomposition and Additive coupling layer. MAGANet incorporates three primary loss functions to train its VAE and flow-based components:

$$
\begin{aligned}
\mathcal{L}_{\text{recon}} &= l_{\mathcal{D}}(D(E(x_1, x_2), x_1), x_2), &(10)\\
\mathcal{L}_{\text{recon\_latent}} &= l_1(E(x, D(z, x)), z), &(11)\\
\mathcal{L}_{\text{Base}} &= \mathcal{L}_{\text{recon}} + \beta_{\text{KL}}\mathcal{L}_{\text{KL}} + \beta_{\text{recon\_latent}}\mathcal{L}_{\text{recon\_latent}}, &(12)
\end{aligned}
$$

where $l_{\mathcal{D}}$ denotes the loss in the image space, $l_1$ represents the $L_1$ norm, $E$ is the encoder, and $D$ is the decoder. The hyperparameters $\beta_{\text{KL}}$ and $\beta_{\text{recon\_latent}}$ control the weighting of the KL divergence and latent reconstruction losses, respectively.

### C.2 PROPOSED METHOD

To generate explicit curve and and approximate geodesic, we use the curve function maker. The structure of module is as followed:

where $h$ is the latent dimension of encoder before sampling layer. After approximate a curve $\gamma'(t)$, we can obtain result by $\gamma(t) = \mu(\gamma'(t)) + \epsilon \times \sigma^2(\gamma'(t))$, where $\mu$ is the mean sampling layer, $\epsilon$ is noise for reparameterization trick and $\sigma^2$ is the log-variance sampling layer.

Table 3: Curve Function Maker (Fully connected layer is denoted in FC)

| Encoder | | | |
|---|---|---|---|
| FC(64) | | | |
| ReLU | | | |
| FC(128) | | | |
| ReLU | | | |
| $W_1$ | $W_2$ | $b_1$ | $b_2$ |
| FC(64) | FC($64 \times h$) | FC(64) | FC($h$) |

## D  EXPERIMENTS DETAILS

### D.1  EXPERIMENT ON MORPHO-MNIST

The Morpho-MNIST dataset (Castro et al., 2019) is a variant of MNIST dataset which has additional label about thickness or deformation of a digit. To split dataset similar to Recombination-to-Range setting, we use thick zero digits as test set and the other combinations of digit and thickness label as train set. The structure of the encoder of VAE is depicted in Table 4. We set the latent vector dimen-

Table 4: VAE encoder for Morpho-MNIST.

| |
|---|
| Convolution layer with 16 channels |
| ReLU |
| Convolution layer with 32 channels |
| ReLU |
| Convolution lyaer with 32 channels |
| ReLU |
| Two fully connected layer with 6 nodes for mean and standard deviation |

sion as six and pick two dimension with the highest Kullback-Leibler divergence for visualization. The structure of the decoder of VAE is transpose of the encoder. The structure of the curve function maker is smaller than the one working with MAGANet and it is depicted in Table 5. The learning

Table 5: Curve Function Maker for Morpho-MNIST (Fully connected layer is denoted in FC)

| Encoder | | | |
|---|---|---|---|
| FC(64) | | | |
| ReLU | | | |
| FC(32) | | | |
| ReLU | | | |
| $W_1$ | $W_2$ | $b_1$ | $b_2$ |
| FC(32) | FC($32 \times 32$) | FC(32) | FC(32) |

rate for vanilla VAE, stand-alone curve function maker and VAE with geodesic symmetry is set as 0.0005 for each model. Each VAE model is trained for 500 epochs and stand-alone curve function maker is trained for 500 epochs.

### D.2  EXPERIMENT ON BENCHMARKS OF COMBINATORIAL GENERALIZATION

**Dataset Setting**   For split dSprites dataset (Matthey et al., 2017) in Recombination-to-Elements setting, we except following combinations:

1. shape=ellipse, scale=0.5, $120° \leq$ orientation $\leq 240°$, $0.6 < x$, $0.6 < y$,

2. scale=0.5, orientation=$0°$, $x \leq 0.25$, $y \leq 0.25$,

3. shape=heart, orientation=$0°$, $0.5 < x$, $0.5 < y$.

In Recombination-to-Range setting, we except following combinations:

1. shape=heart, $0.5 < x$,

2. shape=square, $0.5 < x$,

3. shape=ellipse, $3 <$ scale, $y < 0.5$.

For 3D Shapes dataset Burgess & Kim (2018) in Recombination-to-Elements setting, we except following combinations:

1. floor-hue$> 0.5$, wall-hue$> 0.5$, object-hue$> 0.5$, scale=7, shape=cube, orientation=$0°$,

2. floor-hue$\leq 0.5$, wall-hue$\leq 0.5$, object-hue$\leq 0.5$, scale=7, shape=cylinder, orientation=$0°$,

3. floor-hue$\leq 0.5$, wall-hue$> 0.5$, object-hue=0, scale=0, shape=[sphere, cube], orientation=$-30°$.

In Recombination-to-Range setting, we except following combinations:

1. $0 \leq$ floor-hue$\leq 1$, $0 \leq$ wall-hue$\leq 1$, object-hue$> 0.5$, $0 \leq$ scale$\leq 1$, shape=oblong, $-30° \leq$ orientation$\leq 30°$,

2. $0 \leq$ floor-hue$\leq 1$, $0 \leq$ wall-hue$\leq 1$, $0 \leq$ object-hue$\leq 1$, scale$\leq 2$, shape=sphere, $-30° \leq$ orientation$\leq 30°$,

3. floor-hue$< 0.5$, $0 \leq$ wall-hue$\leq 1$, $0 \leq$ object-hue$\leq 1$, $0 \leq$ scale$\leq 8$, shape=cylinder, $-30° \leq$ orientation$\leq 0°$.

**Computing Resource**   We conducted experiments on local server equipped with NVIDIA graphic card RTX 2080Ti, RTX 3090 or A100. Each run requires approximately 6000MiB of VRAM and takes about 30 hours. These requirements may vary depending on the dataset, split settings, and GPU used.

