# OpenReview forum: "Symmetric Space Learning for Combinatorial Generalization"
_ICLR.cc/2025/Conference — Submitted to ICLR 2025_

### Official Review · Reviewer_Cbnk · 2024-10-20

**Soundness:** 2
**Presentation:** 2
**Contribution:** 1
**Rating:** 3
**Confidence:** 3

**Summary:**

This paper studies the symmetry generalization problem in the latent space of a dataset. Specifically, the authors assume the data manifold and its latent space are homogeneous with some group actions, then the unseen samples can be generalized by some group actions on the existing samples as the group is assumed to act transitively.

**Strengths:**

The authors propose a novel approach through learning the symmetry in the latent space to solve the combinatorial generalization problem in the data space.

**Weaknesses:**

1. The paper is not carefully written with several grammatical errors. For example, the first sentence of Proposition 2; between line 172-173, "However, encoding data onto a latent manifold enables data to be handled in lower dimensional and easy-to-compute data."

2. Condition 3 is the key assumption of this paper, but it is too strong without any justification.

3. I'm confused by the last sentence between line 174-175, is $\mathcal{N}\subset\mathcal{Z}$ or vice versa?

4. The math part is presented in a redundant way. I think the authors can make it more clear by clarifying all the assumptions at once. Also, the propositions and theorems are trivial, and they are essentially the mathematical formulations of the assumptions.

5. Condition 4 is also a very strong assumption, and it requires $\mathcal{M}$ to have the same dimension as $\mathcal{N}$, so why is $\phi$ an encoder?

**Questions:**

See Weaknesses above.

---

> ### Author Response · Authors · 2024-11-26
>
> Thank you for your detailed reading and thoughtful review of our work. Your feedback has been invaluable in improving our research. Below, we respond to the concerns raised, particularly concerning grammatical errors and the mathematical sections, which have undergone significant revision. We kindly ask you to review the revised version, where major changes are highlighted in yellow and red for clarity.
>
> ### Weaknesses:
>
> - W1 The paper is not carefully written with several grammatical errors. For example, the first sentence of Proposition 2; between line 172-173, "However, encoding data onto a latent manifold enables data to be handled in lower dimensional and easy-to-compute data."
>     - A1 We have revised the entire paper to correct grammatical errors and enhance coherence.
> - W2 Condition 3 is the key assumption of this paper, but it is too strong without any justification.
>     - A2 Condition 3 describes the state of the ideal solution to the problem addressed in Proposition 2. Our method does not rely on this assumption prior to application but instead seeks to induce the condition through a practical approach. In Subsection 6.1 and Figure 3, we demonstrate that our model can uncover geometric information and transfer local symmetries within the latent space of the VAE model along approximated geodesics. Additionally, the experimental results presented in Subsection 6.2 validate the effectiveness of our method. Notably, previous studies have also utilized homogeneous spaces [1, 2] and Lie groups [3] as symmetries acting on a manifold, yielding similarly positive effects.
> - W3 I'm confused by the last sentence between line 174-175, is $\mathcal{N}\subset\mathcal{Z}$ or vice versa?
>     - A3 The terms written in **mathcal** refer to spaces, while those written in normal font refer to sets. In the sentence, we have two terms that may cause visual confusion; the latent space $\mathcal{Z}$ and the set of latent vectors $Z$. Thus, $Z\subset\mathcal{N}\subset\mathcal{Z}$. To reduce this confusion, we revise the notation: $\mathcal{D}$ denotes the data space which was denoted as $\mathcal{X}$, and $\mathcal{H}$ denotes the latent space which was denoted as $\mathcal{Z}$.
> - W4 The math part is presented in a redundant way. I think the authors can make it more clear by clarifying all the assumptions at once. Also, the propositions and theorems are trivial, and they are essentially the mathematical formulations of the assumptions.
>     - A4 The propositions establish the conditions for applying our method and explain how it serves as a solution. We simplify and revise the mathematical section accordingly to clarify their roles and reduce redundancy.
> - W5 Condition 4 is also a very strong assumption, and it requires $\mathcal{M}$ to have the same dimension as $\mathcal{N}$, so why is $\phi$ an encoder?
>     - A5 Our aim is to use MAGANet as the base architecture, which incorporates a GLOW module to meet the invertible, differentiable, and diffeomorphic conditions, as demonstrated in [4]. To avoid misleading readers regarding its generality, we have replaced the term “encoder” with "map" throughout the paper.
>
> [1] Cohen, Taco S., Mario Geiger, and Maurice Weiler. "A general theory of equivariant cnns on homogeneous spaces." *Advances in neural information processing systems* 32 (2019).
>
> [2] Moriakov, Nikita, Jonas Adler, and Jonas Teuwen. "Kernel of CycleGAN as a principal homogeneous space." *International Conference on Learning Representations*.
>
> [3] Lu, Mei, and Fanzhang Li. "Survey on lie group machine learning." *Big Data Mining and Analytics* 3.4 (2020): 235-258.
>
> [4]Zhen, Xingjian, et al. "Flow-based generative models for learning manifold to manifold mappings." *Proceedings of the AAAI Conference on Artificial Intelligence*. Vol. 35. No. 12. 2021.

---

> ### Comment · Reviewer_Cbnk · 2024-12-02
>
> Thank you for the response. I am satisfied with your response to W2-W3.
>
> However, my concerns in W4 and W5 remain:
>
> W4 -- The theory is not novel and basically formulates the assumptions. I don't see any theoretical insights for the proposed Geodesic Symmetries.
>
> W5 -- I think the invertible, differentiable, and diffeomorphic conditions are too strong. For example, the approach in a recent paper [1] seems more general than your assumptions.
>
> Given some of my concerns have not been addressed, I would like to maintain my score.
>
> [1] Jin, Y., Shrivastava, A. and Fletcher, T., Learning Group Actions on Latent Representations. NeurIPS 2024.

---

### Official Review · Reviewer_W48H · 2024-10-29

**Soundness:** 2
**Presentation:** 2
**Contribution:** 2
**Rating:** 5
**Confidence:** 3

**Summary:**

This paper addresses the problem of combinatorial generalization onto unseen data by enforcing the latent space to be a symmetric space. To achieve this, the paper proposes several loss terms to enforce the geodesic symmetry in the learned latent. The method is tested on standard datasets for combinatorial generalization, including dSprites and 3D Shapes.

**Strengths:**

The paper discusses an interesting topic, combinatorial generalization using symmetry-based representation learning.

**Weaknesses:**

The paper is poorly written. It contains many grammatical errors, confusing phrases, and informal statements, and it lacks coherence overall. As a result, the goal and the contribution of this paper are totally unclear to me, which is my main concern before all others. As far as I can understand, the authors are trying to address the problem in combinatorial generalization that symmetry transformation may take data points outside the training domain. This problem is indeed valid, but I fail to see how the theories in Section 3 or the method in Section 4 can solve it. In particular,
* It seems that the idea of this paper is to learn a latent space which is a homogeneous space by enforcing the model to align with the geodesic symmetry (please correct me if this is wrong). But there is not enough explanation as to why this would work. I notice Thm 3 does mention something related, but it is stated informally and vaguely (e.g. What does it mean by "make symmetric space **enough**"? ) and the proof is missing.
* The method involves approximating the geodesic. It simply parameterizes an arbitrary path in the ambient and penalizes its length in the interval $t \in [0, 1]$ and the reversal symmetry error at a single point $t=-1$. There is no justification for why (1) this would result in a path that stays on the latent manifold (moreover, the latent manifold itself has not been identified anywhere in the discussion), (2) the learned path function would generalize to $t<0$, and (3) the geodesic symmetry would hold for all $t$ instead of only at $t=1$.
* The role and properties of the anchor $a$ are also not adequately explained. It is unclear whether any constraint needs to be enforced for this learnable anchor to make sure its properties match those in the definition of symmetric space.
* I am also confused by the fact that, generally in symmetry-based representation learning, the symmetry transformations represent the factors of variations, and depending on specific data generation procedures there should be different symmetry groups, while this paper seems to only consider the geodesic symmetry as defined in Defn 3.

Some other less major concerns, mainly about presentation and organization of the paper:
* Definitions in Section 2: instead of making a separate paragraph title (e.g. symmetric space), you can specify the terminologies right after the definition header (e.g. Definition 3 (symmetric space)).
* Definition 3: is there a confusion between $o$ and $a$? Should it be $T_a \mathcal M$? Also, what does it mean by "$U$ can be extended to the entire manifold"?
* Section 3 can be made more concise. The proofs are too verbose. Statements like Prop 2 and Thm 1 just directly follow from definitions.
* Thm 1: Let $X, X'$ be partitions of a manifold $\mathcal M$. This statement is misleading. Partition can mean different things according to context. If the authors mean these two sets are disjoint and their union is the entire manifold, a better way to phrase it may be ``$\\{X, X'\\}$ is a partition of $\mathcal M$''.
* The training objective, Eq (9), involves a base loss from another paper called MAGANet. This name came out of nowhere for the first time in L315. The authors should explain this in more detail, or at least reference the original paper here. Also, the authors should discuss whether the proposed method can only be applied to MAGANet, which would be a potential limitation.

**Questions:**

* What is $\epsilon$ in Eq (2)? Is it a random noise? What is the reason for adding a noise component to the path $\gamma$?
* Can the proposed method be applied to models other than MAGANet?
* Why do the results in Table 1 have such large differences? Does a BCE of 3000+ indicate total failure? Also, for the 3D Shapes dataset, $\beta$-VAE ($\beta=2$) seems to perform best in all cases. Why is your method highlighted then?
* In Fig 1 caption: Why is the volume form related to the likelihood?
* Can you elaborate on why you need to exclude three specific combinations of factors?

---

> ### Author Response · Authors · 2024-11-26
>
> Thank you for your detailed and thoughtful review. We regret that the incomplete clarity and coherence of the original draft may have hindered the effective communication of our intended ideas. In response, we have made extensive revisions, including correcting grammatical errors, improving readability, and enhancing coherence. Notably, substantial updates have been made to the mathematical sections. Below, we address your concerns and questions, and we kindly ask you to review the revised version, where significant changes are highlighted in yellow and green for clarity.
>
> ### Weaknesses:
>
> - W1 The paper is poorly written. It contains many grammatical errors, confusing phrases, and informal statements, and it lacks coherence overall. As a result, the goal and the contribution of this paper are totally unclear to me, which is my main concern before all others. As far as I can understand, the authors are trying to address the problem in combinatorial generalization that symmetry transformation may take data points outside the training domain. This problem is indeed valid, but I fail to see how the theories in Section 3 or the method in Section 4 can solve it
>     - A1 To improve clarity, we have revised the paper, mainly to emphasize the role of Section 3 and its connections to Section 4.
> - In particular,
>     - W1.1 It seems that the idea of this paper is to learn a latent space which is a homogeneous space by enforcing the model to align with the geodesic symmetry (please correct me if this is wrong). But there is not enough explanation as to why this would work. I notice Thm 3 does mention something related, but it is stated informally and vaguely (e.g. What does it mean by "make symmetric space enough"? ) and the proof is missing.
>         - A1.1 We revised Thm 3 (**Prop. 3** **in the revised version**) to enhance clarity and **completed its proof**. The core idea is that leveraging homogeneity allows symmetries to extend to unseen regions. Geodesic symmetries and symmetric spaces bridge the gap between homogeneity and its implementation by utilizing geometric information from geodesics. We show that learning geodesic symmetries extends automorphisms in observed regions, overcoming the limitations of previous symmetry-based methods.
>     - W1.2 The method involves approximating the geodesic. It simply parameterizes an arbitrary path in the ambient and penalizes its length in the interval $t\in[0,1]$ and the reversal symmetry error at a single point $t=-1$. There is no justification for why (1) this would result in a path that stays on the latent manifold (moreover, the latent manifold itself has not been identified anywhere in the discussion), (2) the learned path function would generalize to $t<0$, and (3) the geodesic symmetry would hold for all $t$ instead of only at $t=1$.
>         - A1.2 We clarified our assumptions based on the manifold hypothesis, which assumes that a lower-dimensional data manifold exists and is embedded into the latent space via the GLOW module [1]. To ensure the curve remains on the latent manifold, we: (1) share the last layer of the VAE encoder for the curve generator because this layer implicitly approximates latent manifolds for observed inputs, and (2) penalize the curve energy, which is exponentially increased when the curve escape from the manifold. These constraints induce the curve to stay near the manifold. For generalization to $t < 0$, if both $\gamma(1)$ and $\gamma(-1)$ lie in the observed region, the curve generalizes by aligning **intermediate points iteratively** through training.
>     - W1.3 The role and properties of the anchor $\alpha$ are also not adequately explained. It is unclear whether any constraint needs to be enforced for this learnable anchor to make sure its properties match those in the definition of symmetric space.
>         - A1.3 To define geodesic symmetry, we require a reference point $\gamma(0)$ for the geodesic $\gamma$. We introduce a shared learnable parameter, the *anchor*, for simplicity and training stability. We experimented with several anchor configurations, including fixing the anchor at 0, assigning specific anchors to parts of the latent manifold, and using a codebook of anchors. However, these settings either showed no improvement or failed to converge. While improving anchor selection may enhance performance, further exploration is left as future work.

---

> ### Author Response · Authors · 2024-11-26
>
> - W1.4 I am also confused by the fact that, generally in symmetry-based representation learning, the symmetry transformations represent the factors of variations, and depending on specific data generation procedures there should be different symmetry groups, while this paper seems to only consider the geodesic symmetry as defined in Defn 3.
>         - A1.4 Here, the symmetries in “geodesic symmetry” and the symmetries to represent “factors of variations” (referred to as factor symmetry) are distinguished. Our goal is to transfer the factor symmetries over the seen region (referred to as local structures) on the manifold to the unseen region through geodesic symmetry to hold the same local symmetries on the two regions. The current geodesic symmetry used in the paper is simply reflection, but it does not mean only reflection is considered in factor symmetries.
> - W2 Some other less major concerns, mainly about presentation and organization of the paper:
>     - W2.1 Definitions in Section 2: instead of making a separate paragraph title (e.g. symmetric space), you can specify the terminologies right after the definition header (e.g. Definition 3 (symmetric space)).
>         - A2.1 We revised Section 2 as suggested. Definitions now include terminologies directly within the headers (e.g., "Definition 4 (Symmetric Space)"). Please see the green highlights in Section 2 for clarity.
>     - W2.2 Definition 3: is there a confusion between $o$ and $a$? Should it be $T_a\mathcal{M}$? Also, what does it mean by "$U$ can be extended to the entire manifold"?
>         - A2.2 We corrected the notation and clarified ambiguities in Definition 3. The phrase "$U$ can be extended to the entire manifold" refers to generalizing geodesic symmetries, initially defined locally, to the entire manifold. For clarity, we revised the definition to avoid such ambiguities.
>     - W2.3 Section 3 can be made more concise. The proofs are too verbose. Statements like Prop 2 and Thm 1 just directly follow from definitions.
>         - A2.3 We revised Thm. 1 (now integrated into Prop. 3) to explicitly state that $X$ and $X^\prime$ are disjoint subsets of $\mathcal{M}$ whose union forms the entire manifold. Please see the updated Prop. 3 in Section 3.2 for details.
>     - W2.4 Thm 1: Let $X,X^\prime$ be partitions of a manifold $\mathcal{M}$. This statement is misleading. Partition can mean different things according to context. If the authors mean these two sets are disjoint and their union is the entire manifold, a better way to phrase it may be “$\{X,X^\prime\}$ is a partition of $\mathcal{M}$''.
>         - A2.4 While we revise Section 3, we fuse Thm 1 into Prop 3 (Thm 2 in the original version). We clarify that what are $X$ and $X^\prime$, and they are partitions of $\mathcal{M}$. Please check the Prop. 3 in Section 3.2.
>     - W2.5 The training objective, Eq (9), involves a base loss from another paper called MAGANet. This name came out of nowhere for the first time in L315. The authors should explain this in more detail, or at least reference the original paper here. Also, the authors should discuss whether the proposed method can only be applied to MAGANet, which would be a potential limitation.
>         - A2.5 We added Subsection 4.1 to the Method section to describe MAGANet and its relevance to our approach. MAGANet was chosen because it satisfies the required conditions, such as equivariant diffeomorphism, through its GLOW module. While our method currently relies on MAGANet, adapting it to other models with relaxed conditions is an open research problem and a potential avenue for future work.

---

> ### Author Response · Authors · 2024-11-26
>
> ### Questions:
>
> - Q1 What is $\epsilon$ in Eq (2)? Is it a random noise? What is the reason for adding a noise component to the path $\gamma$?
>     - QA1 The $\epsilon$ term is derived from the VAE encoder[2] and represents variability in the latent space. By including this component, the curve remains within the learned marginal distribution of the encoder, helping it stay close to the latent manifold, as explained in A1.2.
> - Q2 Can the proposed method be applied to models other than MAGANet?
>     - QA2 In principle, any model satisfying the required conditions (e.g., equivariant diffeomorphism) can adopt our method. In this work, we focus on demonstrating the effectiveness of symmetric space using MAGANet, a model that practically satisfies these conditions, leaving exploring extensions to other models as future work.
> - Q3 Why do the results in Table 1 have such large differences? Does a BCE of 3000+ indicate total failure? Also, for the 3D Shapes dataset, $\beta$-VAE $\beta=2$ seems to perform best in all cases. Why is your method highlighted then?
>     - QA3 There was an error in Table 1; the $\beta=2$ rows were swapped. The corrected table is now provided with green highlights. Please see the updated Table 1.
> - Q4 In Fig 1 caption: Why is the volume form related to the likelihood?
>     - QA4 The volume form relates to the Riemannian metric on the latent manifold, which determines sensitivity to factor changes. Similar usage of volume forms can be found in [3,4].
> - Q5 Can you elaborate on why you need to exclude three specific combinations of factors?
>     - QA5 We designed three settings to evaluate robustness systematically. The first matches previous works, the second contrasts this setup, and the third excludes specific ranges. These settings provide a balanced and comprehensive analysis of factor combination robustness.
>
> [1] Kingma, Durk P., and Prafulla Dhariwal. "Glow: Generative flow with invertible 1x1 convolutions." *Advances in neural information processing systems* 31 (2018).
>
> [2] Burgess, Christopher P., et al. "Understanding disentangling in $\beta $-VAE." *arXiv preprint arXiv:1804.03599* (2018).
>
> [3] Chadebec, Clément, and Stéphanie Allassonnière. "A geometric perspective on variational autoencoders." *Advances in Neural Information Processing Systems* 35 (2022): 19618-19630.
>
> [4]Chen, Nutan, et al. "Metrics for deep generative models." *International Conference on Artificial Intelligence and Statistics*. PMLR, 2018.

---

> > ### Comment · Reviewer_W48H · 2024-12-02
> >
> > I thank the authors for their detailed response and efforts to revise the paper. The revised manuscript is much clearer to me. Since many of my questions have been addressed, I will increase my score to 5. I will not raise my score higher because there are still some remaining concerns, which I hope the authors could consider in future revisions.
> >
> > * Double-check the correctness of the theorems. For example, Proposition 1 seems problematic - if G acts transitively on the entire space, how can there exist no group element that transforms x to x'?
> > * Also, try to avoid ambiguity in theorems. For example, in Prop 3, "the model learns Aut(Z)" is confusing. What is "the model"? In fact, when you make such a mathematical statement, you don't have to specify your implementation of it. "There exists $\alpha \in \mathrm{Aut}(Z)$" would be enough. You can explain your methodology and machine learning model later in the text. Moreover, the notation $G/H_a$ was not defined anywhere before. I assume it should have been covered in the definition of symmetric space.
> > * As also pointed out by other reviewers, the proposed solution, while effective, can only be applied under some nontrivial conditions. Besides, the current method has many trainable modules and multiple training objectives, and I feel more ablation studies should be performed to identify which component can effectively improve performance.

---

### Official Review · Reviewer_dKPg · 2024-11-04

**Soundness:** 3
**Presentation:** 3
**Contribution:** 3
**Rating:** 6
**Confidence:** 4

**Summary:**

This paper proposes a novel method for symmetry generalization -extending trained symmetries on observed data to unseen data for combinatorial generalization. In this approach, the latent space manifold is enforced to be a symmetric space, and homogeneous conditions on the data manifold as well as the latent manifold are present. The sampling method proposed here explicitly learns geodesic symmetry, which helps reduce the computational costs for sampling and induces symmetric space for latent manifolds.
The paper presents empirical experiments through qualitative and quantitative analysis on datasets like MNIST, dSprites and 3D-Shapes.

**Strengths:**

1. The paper presents a well-defined idea to build a non-rigid approach to building symmetries and uses this for combinatorial generalization.

2. The proposed method builds a local structure in the latent manifold which can therefore be used for geodesic symmetries. I believe this approach can be useful for different tasks like latent disentanglement.

**Weaknesses:**

1. The proposed method uses one object per data sample, be it MNIST, dSprites, or 3Dshapes, such that the geodesic symmetry approach works with any issues of entanglement. It would be interesting to show how this extends in multi-objects set-up.

2. The homogeneity conditions proposed for both data manifold and latent manifold, might not strictly hold in the case of 2 or more objects in a sample.

**Questions:**

1. How does strict permutation invariance in latent generative models compare to this approach?
2. Previous works like [1,2] are capable of representing invariant shapes as latent. How does this explicit and exact symmetry equivariant VAEs compare to this work?
3. Can this approach be extended to hierarchical latent models like NVAE?



[1] Continuous Kendall Shape Space VAEs, Vadgama et al

[2] EqVAE:Equivariant Priors for Compressed Sensing with Unknown Orientation. Kuzina et al.

---

> ### Author Response · Authors · 2024-11-26
>
> Thank you for your insightful review and for suggesting potential extensions, particularly regarding the multi-object environment, which we find to be a fascinating and valuable direction for future research. Below, we respond to your concerns and questions. We have made significant revisions to the paper and kindly ask you to review the updated version. Fundamental changes are highlighted in yellow for your convenience. Should you have any additional questions, please feel free to share them with us.
>
> ### Weaknesses:
>
> - W1: The proposed method uses one object per data sample, be it MNIST, dSprites, or 3Dshapes, such that the geodesic symmetry approach works with any issues of entanglement. It would be interesting to show how this extends in multi-objects set-up.
>     - A1: Extending symmetry-based methods to multi-object environments is indeed a compelling and challenging problem. However, a consensus theoretical definition for symmetries in multi-object settings is still lacking. In single-object environments, each latent vector represents one object and its attributes, which can be expressed as symmetries. In multi-object environments, latent vectors must encode objects, their attributes, and their interrelationships. Modeling relationships, especially asymmetric ones like spatial relationships, presents significant difficulties for symmetry-based methods. Extending our approach to multi-object settings is an important avenue for future research.
>
> - W2: The homogeneity conditions proposed for both data manifold and latent manifold, might not strictly hold in the case of 2 or more objects in a sample.
>     - A2: Our current focus is on single-object environments to establish foundational insights. Generalizing these conditions to multi-objective settings, potentially with relaxed or modified homogeneity requirements, remains an open research direction.
>
> ### Questions:
>
> - Q1: How does strict permutation invariance in latent generative models compare to this approach?
>     - QA1: Permutation invariance can facilitate combinatorial generalization by decomposing objects into attributes. However, our approach addresses the combinatorial generalization task through symmetry generalization. This method may have advantages in scenarios with little to no prior knowledge. Conversely, it might be less effective when prior knowledge is present, and the data can be explicitly decomposed into attributes.
> - Q2: Previous works like [1,2] are capable of representing invariant shapes as latent. How does this explicit and exact symmetry equivariant VAEs compare to this work?
>     - QA2: In combinatorial generalization tasks, our method avoids leveraging prior knowledge about factors, enabling generalization in an unsupervised manner. In contrast, prior works assume known symmetries, such as rotation invariance, during training. Our method focuses on discovering and generalizing symmetries without relying on pre-defined invariances.
> - Q3: Can this approach be extended to hierarchical latent models like NVAE?
>     - QA3: Directly applying this method to hierarchical models like NVAE poses challenges since NVAE does not explicitly model symmetries and needs a diffeomorphic mapping. While integrating symmetry learning into hierarchical models is an interesting direction, it is beyond the scope of our current work.

---

### Official Review · Reviewer_yCwt · 2024-11-05

**Soundness:** 4
**Presentation:** 3
**Contribution:** 2
**Rating:** 5
**Confidence:** 2

**Summary:**

The authors propose a method to enforce the latent space to be symmetric in order to generalize combinatorially to symmetries. By first learning the symmetric latent space from training data, the network can generalize to combinations of symmetries at inference. Specifically, they learn the geodesics of the latent manifold and extrapolate points on the geodesic by reflection. The authors provide theoretical results on how enforcing homogeneity and isometry on the latent space and using an equivariant map induces the data manifold to also be symmetric. Experiments on MorphoMNIST and dSprites show better results than baselines.

**Strengths:**

- The authors clearly define a symmetric space and provide proofs on how properties of the symmetric space lend themselves to unseen combinations of symmetries
- The practical implementation seems relatively straightforward and is clearly described.
- The experiments show favorable results compared to MAGANet and BetaVAE.

**Weaknesses:**

- The theoretical results seem somewhat obvious and have been shown before [1]. Furthermore, there seems to be a disconnect between the theory and the practical implementation. Could the authors clarify how enforcing the various loss terms ensures that the encoder is an equivariant diffeomorphism?
- The experiments were done on somewhat toy domains. In particular, I feel that the 3D-Shape dataset shows the objects in different colors relative to the background, making the task easier. What would happen if the 3D-Shapes dataset did not include hue?
- Baselines: MAGANet seems to be the only baseline that aims to learn the symmetries directly. It would have been nice to consider other methods that learn symmetries [2], [3]

References:
1. Cohen, T. S., Geiger, M., & Weiler, M. (2019). A general theory of equivariant cnns on homogeneous spaces. Advances in neural information processing systems, 32.
2. Falorsi, L., De Haan, P., Davidson, T. R., De Cao, N., Weiler, M., Forré, P., & Cohen, T. S. (2018). Explorations in homeomorphic variational auto-encoding. arXiv preprint arXiv:1807.04689.
3. Miyato, T., Koyama, M., & Fukumizu, K. (2022). Unsupervised learning of equivariant structure from sequences. Advances in Neural Information Processing Systems, 35, 768-781.

**Questions:**

- Figure 3: are the black lines the true geodesic? How were they computed?
- How do you find the anchor for geodesics?

---

> ### Author Response · Authors · 2024-11-26
>
> Thank you for your detailed review and thoughtful comments, particularly regarding the experimental settings and results. We have addressed your concerns in our responses below. Additionally, we have significantly revised the paper by improving the theoretical sections and adding new baselines. We kindly request that you review the updated version, focusing on the highlights in yellow and cyan for the key changes.
>
> ### Weaknesses:
>
> - W1: The theoretical results seem somewhat obvious and have been shown before [1]. Furthermore, there seems to be a disconnect between the theory and the practical implementation. Could the authors clarify how enforcing the various loss terms ensures that the encoder is an equivariant diffeomorphism?
>     - A1: While prior works, including [1], on homogeneous spaces assume full data observability, we focus on generating new combinations from partial observations. This problem has yet to be explored with homogeneous spaces so far. Our contributions demonstrate that symmetric spaces serve as a key solution. Regarding the encoder, we clarify that the forward process of the GLOW module in MAGANet is equivariant and invertible, in Section 4.1. No additional loss term is needed for equivariant diffeomorphism, as the GLOW module inherently satisfies these properties.
> - W2: The experiments were done on somewhat toy domains. In particular, I feel that the 3D-Shape dataset shows the objects in different colors relative to the background, making the task easier. What would happen if the 3D-Shapes dataset did not include hue?
>
>     - A2: The application of homogeneous spaces for generalization remains in its early, theoretical stages, as enforcing the required conditions across the entire space presents significant practical challenges. In this work, we focus on demonstrating the potential of symmetric spaces for generalization rather than their full applicability in practical settings. In the case of 3D Shapes, if the 3D Shapes dataset did not exclude combinations with ranges of hue (if this is not what you meant, please let us know), the task would become easier. Factors other than hue have fewer elements, resulting in smaller unseen regions. Our data settings already include combinations where the object and background share the same hue. Please refer to Appendix D.2 for further details.
>
> - W3: Baselines: MAGANet seems to be the only baseline that aims to learn the symmetries directly. It would have been nice to consider other methods that learn symmetries [2], [3]
>     - A3: We chose MAGANet because it satisfies the equivariant diffeomorphism condition through its GLOW module, which aligns with the requirements of our method. However, we also conducted experiments with CLGVAE [4], which did not produce competitive results. Please see the cyan-highlighted rows in Table 1 for the performance of CLGVAE. The result of CLGVAE:
>
> |  |  | R2E |  |  | R2R |  |
> | --- | --- | --- | --- | --- | --- | --- |
> |  | case1 | case2 | case3 | case1 | case2 | case3 |
> | dSprites | 9.69 | 9.34 | 16.24 | 448.28 | 343.98 | 66.04 |
> | 3D Shapes | 7428.74 | 3624.50 | 3952.53 | 3862.72 | 3766.28 | 3740.66 |
>
> ### Questions:
>
> - Q1: Figure 3: are the black lines the true geodesic? How were they computed?
>     - A1: The black lines are linear extrapolations of each vector about the anchor. They are used to compare approximate geodesics.
> - Q2: How do you find the anchor for geodesics?
>     - A2: The anchor is a learnable parameter optimized through Eq. (9). Practically, it is implemented using the **torch.nn.Parameter** class. We experimented with various anchor configurations, but the learnable anchor setting yielded the best results.
>
> [4] Zhu, Xinqi, Chang Xu, and Dacheng Tao. "Commutative lie group vae for disentanglement learning." *International Conference on Machine Learning*. PMLR, 2021.

---

> ### Comment · Reviewer_yCwt · 2024-11-27
>
> I thank the authors for their explanations. The explanation for the encoder raised another question for me: I get that GLOW is a diffeomorphism, but why is it equivariant? As far as I know, GLOW does not have any mechanisms to respect the group action.

---

> > ### Author Response · Authors · 2024-11-27
> >
> > While equivariance cannot be defined on for GLOW standalone, original MAGANet (page 5 in [5]) is explicitly designed to be equivariant. Specifically, the following equations are held:
> >
> > $$
> > x^\prime_2=\phi(x_2)=\phi(x_1\cdot z)
> > $$
> > $$
> > x^\prime_2=\rho_\mathcal{H}(z)(x^\prime_1)=\rho_\mathcal{H}(z)\circ\phi(x_1)
> > $$
> >
> > where $x_1,x_2\in \mathcal{D}$, $x^\prime_1,x^\prime_2\in\mathcal{H}$ and $\cdot z$ denotes a group action from $x_1$ to  $x_2$, $\rho_\mathcal{H}(z)$ represents a group action from $x^\prime_1$ to $x^\prime_2$. According to MAGANet paper, the decoder of MAGANet has following structure (adapted to our notation):
> >
> > $$
> > D(z,x)=\phi^{-1}\circ\rho_\mathcal{H}(z)\circ\phi(x).
> > $$
> >
> > This structure naturally becomes equivariant due to the invertibility of the mapping between the data space and the latent space. Our approach inherits this decoder structure. Consequently, the MAGANet decoder, composed of GLOW and the modeled group action, ensures equivariance.
> >
> > [5] Hwang, Geonho, et al. "MAGANet: Achieving combinatorial generalization by modeling a group action." *International Conference on Machine Learning*. PMLR, 2023.

---

### Meta-Review · Area_Chair_sPMA · 2024-12-18

**Metareview:**

The work claims that symmetric latent spaces can address the problem of combinatorial generalization, and several loss terms are proposed which enforce latent symmetry. Improvements on dSprites and 3D Shapes datasets are claimed.

While the presented approach on learning symmetric latents using geodesics is interesting, there are several issues with the paper such as correctness of Proposition 1 and 3 as pointed out by reviewer W48H. While the proposed algorithm and framework is interesting, the numerical results are done on toy domains (dSprites and 3D Shape) and larger scale evaluations would make the work more convincing. At this stage the paper does not meet the standards for acceptance at ICLR and encourage the authors to take the reviewers feedback into account for a resubmission.

**Additional Comments On Reviewer Discussion:**

Reviewer W48H and Reviewer Cbnk raised several issues regarding writing, clarity and novelty of the paper which were not fully addressed even after the rebuttal reflected in their scores.

Reviewer yCwt also finds that the contributions are not novel in view of related work and comparisons are missing.

The above points equally influenced my decision.

---

### Decision · Program_Chairs · 2025-01-22

Reject